# Literature Review of the Principal Diagnostic Tests to Detect Bovine Respiratory Disease in Pre-Weaned Dairy and Veal Calves

**DOI:** 10.3390/ani14020329

**Published:** 2024-01-21

**Authors:** Julie Berman

**Affiliations:** Département des Sciences Cliniques, Faculté de Médecine Vétérinaire, Université de Montréal, Saint-Hyacinthe, QC J2S 2M2, Canada; julie.berman@umontreal.ca

**Keywords:** infectious bronchopneumonia, sensitivity, specificity, cattle

## Abstract

**Simple Summary:**

Bovine respiratory disease is a viral or bacterial infection of the upper and lower respiratory tract. This disease is common in pre-weaned dairy and veal calves. This review reports on the technique, interpretation and performance of the different diagnostic tests that can be used in the field to detect bovine respiratory disease.

**Abstract:**

Bovine respiratory disease (BRD) is an infection of the upper and lower respiratory tract, characterized by an inflammation of the lung. Different diagnostic tests can be used to detect BRD, including clinical respiratory scoring systems, thoracic auscultation, and imaging tests like thoracic ultrasonography and thoracic radiography. Although commonly used, none of these diagnostic tests are perfect for detecting BRD. This article reviews the advantages and drawbacks of these techniques and their performance in detecting BRD in pre-weaned dairy and veal calves.

## 1. Introduction

Bovine respiratory disease (BRD) is a viral or bacterial infection of the upper and lower respiratory tract [1]. Infectious bronchopneumonia (BP) is included in the term BRD and is the infection of the lower respiratory tract, characterized by inflammation of the lung [1]. Lower respiratory tract infection is mainly treated with antimicrobials, in contrast with upper respiratory tract infection, which can only be treated with anti-inflammatories [2]. It is, therefore, more useful and accurate to detect BP to guide antimicrobial treatment [2]. 

Multiple factors can be responsible for the occurrence of BRD, such as stress factors (e.g., transport, medical procedures, commingling), environmental factors (e.g., ventilation, season, humidity), immunity (e.g., failure of passive transfer of immunity, host tolerance), and the virulence of infectious agents (e.g., type, strains) [3,4]. Because of these multifactorial characteristics, and depending on the pathogen agent that is involved, the individual and herd expression of BRD varies, making diagnosis challenging [5]. 

The BRD prevalence in pre-weaned dairy calves varies from 12% to 23% in the dairy industry [6,7,8], and from 14% to 61% in veal calves [9,10,11]. In addition to being prevalent, BRD causes major long- and short-term economic losses, such as mortality, decreased growth, reproductive performance, and milk production [12,13,14]. To limit these consequences, the high consumption of antibiotics is commonly used to try to prevent and control BP in pre-weaned calves [15,16,17], which contributes to the potential emergence of resistance [18,19]. The prevention and both the early detection and treatment of BP are essential ways to reduce both economic losses and antibiotic consumption. 

Different diagnostic tests are available to detect BRD and, more accurately, BP. In practice, the most popular diagnostic techniques are clinical respiratory scoring systems (CRSCs), thoracic auscultation (AUSC), and imaging tests like thoracic ultrasonography (TUS) and thoracic radiography (TR) [2]. However, all these diagnostic tests are imperfect, meaning that their sensitivity (Se, i.e., their ability to detect sick calves), specificity (Sp, ability to detect healthy calves), and the agreement between operators are not optimal. Knowledge of those performance limits is important to optimize their sequential use and improve BRD detection. 

When we interpret the Se and Sp of a test in a diagnostic study, it is important to assess potential biases that can affect the exactitude of the results [2]. The frequent biases occurring during a diagnostic study are spectral bias, occurring when the study population does not include the complete spectrum of the disease (e.g., case-control study, referral population); classification bias, occurring when the comparator test is not a 100% accurate gold standard; and incorporation bias, occurring when the results of the investigated test are used to confirm the disease [2,20]. In addition to the exactitude of the results, precision around Se and Sp is also important to assess (e.g., 95% Bayesian credible interval (95%BCI) or 95% confidence interval (95%CI)). Finally, it is also important to consider which status of the disease is measured, such as both lower and upper infection (BRD), active BP (BP with inflammation and infection which needs treatment), inactive BP (where both inflammation and infection are no longer present), clinical BRD, subclinical BRD, etc. 

The aim of this study was, therefore, to review the techniques of CRSC, AUS, TUS, and TR, their performance (Se, Sp, agreement between operators), and their limits of estimation (biases, precision, status investigated) to detect BRD in pre-weaned dairy and veal calves.

## 2. Clinical Respiratory Scoring Systems

The clinical expression of BRD in pre-weaned dairy and veal calves is variable within and between calves [4,21]. Moreover, both the observation and interpretation of BRD clinical signs are difficult and depend on the observers [22]. Clinical respiratory scoring systems have been created to improve the objectivity of both the observation and interpretation of these BRD clinical signs. A clinical scoring system consists of adding weighted predictors when they are present, BRD clinical signs in the case of CRSCs, and attributing a final score. If this final score is greater than or equal to a defined threshold, the CRSC is positive; otherwise, it is negative [23,24]. Due to their simplicity, CRSCs are helpful tools for producers to detect BRD. 

### 2.1. Clinical Respiratory Scoring Systems Used in Pre-Weaned Dairy and Veal Calves

Table 1 shows the CRSCs used in pre-weaned dairy and veal calves. The major drawbacks of those CRSCs are their lack of statistical development (e.g., Wisconsin score) and internal and external validation (e.g., California score) [24]. That is why the optimal threshold is disputable and sometimes changes according to the studies (Wisconsin score ≥ 6 [25], vs. ≥5 [26]) and populations (California score vs. California score from Québec). Finally, only the Berman score was developed and validated to specifically detect active BP and guide producers in establishing accurate BP treatments. 

### 2.2. Performance of Clinical Respiratory Scoring Systems Used in Pre-Weaned Dairy and Veal Calves

Table 2 shows the Se and Sp of the CRSCs used in pre-weaned dairy and veal calves. The limits of each diagnostic study are specified in the last column. All CRSCs reported their performance in detection calves with BRD, i.e., upper infection and active or inactive BP, except one, the Berman score, which focuses on active BP. 

Globally, if we calculate the mean of both the inferior and superior limits of CRSCs used in pre-weaned dairy and veal calves to detect BRD, we obtain a mean Se between 30% and 72% and a mean Sp between 86% and 94%. In practice, if we use the CRSCs in a population with a BRD prevalence of 20%, between 30% and 70% of sick calves would not be diagnosed with BRD (false negatives). At the same time, between 6% and 14% would be diagnosed and probably treated unnecessarily with antibiotics (false positives). 

Only one score (Berman score) has been developed and validated to detect active BP at the group level in veal calves [29]. The score shows that a batch with ≥3 positive-scoring calves among 10 calves, sampled 2 weeks after arrival at the fattening unit, had a 94% chance of having an active BP prevalence ≥ 10%. A batch with <3 positive calves had a 95% chance of not having an active BP prevalence ≥ 10% [29]. The results are promising in guiding the instauration of group treatments according to the active BP prevalence in a batch. Objective measures (e.g., impact on mortality rates, morbidity rates, and number of antimicrobial treatments during the fattening period) are needed in the future to judge this score’s relevance in veal calves.

### 2.3. Agreement between Operators of Clinical Respiratory Scoring Systems

Agreement between operators was only reported for the Wisconsin score at various thresholds (≥4, ≥5, or ≥6) [34]. Whatever the threshold, the agreement was weak when the score was used by three veterinarians (indicators of agreement < 0.4). The results of both the California and Wisconsin scores show perfect agreement when they were used by one operator, an experimented veterinarian [35]. The Berman score includes only clinical signs that are repeatable between different operators (veterinarians, technicians, and producers) (agreement indicators ≥ 0.6), but the interrater agreement between the global score was not determined [22]. 

Clinically, a suboptimal agreement means that two different veterinarians will not classify a calf similarly. At the group level, if a CRSC is used to treat calves for BRD, that implies that two batches or groups with the same BRD prevalence can have a different proportion of treatments. Interrater agreement needs to be reported and improved for each CRSC. Additionally, the agreement was estimated between veterinarians. We do not know the agreement between the producers who use those CRSCs the most. 

Conclusion: Clinical respiratory scoring systems are simple tests that are used for producers to detect BRD in pre-weaned dairy and veal calves. However, there is currently no perfect CRSC. Indeed, CRSCs’ performance is suboptimal in accurately detecting individual sick and healthy calves. Moreover, there is variability in their use between operators, making the results of these tests variable. Currently, other diagnostic tests need to be used after CRSCs to detect BRD and active BP, which need treatment, more accurately. The use of CRSC at the group level (Berman score) to guide group treatment is promising but needs to be justified with objective measures (decreased mortality and morbidity rates or the consumption of antibiotics).

## 3. Thoracic Auscultation

Thoracic auscultation is defined as the science and art of listening to and interpreting sounds from the lungs and respiratory tract [36]. This test needs a stethoscope to transmit sounds from the thoracic cavity to the ears. Thoracic auscultation is simple, fast, and requires limited materials. Practitioners have performed this test for many years during physical exams [36].

### 3.1. Technique of Thoracic Auscultation

The stethoscope is positioned on the whole pulmonary area on both sides (Figure 1). The location of abnormal breath sounds detected with the stethoscope is not the location of the lesions because of the sounds’ irradiation [32,37]. Abnormal breath sounds are, however, rarely audible in the dorsal lung portions with BP [38]. The auscultation area could therefore be limited to the cranial and medial lung portions so that it is carried out faster when multiple calves need to be auscultated (Figure 1) [38]. A bag test can be performed to increase respiratory amplitude. A sleeve glove is put on the calf’s nose. By breathing through the glove, the inhaled air quantity is limited, increasing the respiratory amplitude and the air turbulence [39]. The bag test is often used to improve the auscultation of abnormal breath sounds in horses [39]. In calves, the benefit of this test is unclear, and it is not often used.

### 3.2. Interpretation of Thoracic Auscultation

The breath sounds detectable by AUS are reported in Table 3. The sounds occurring during BP are increased bronchial sounds, crackles, wheezes, pleural friction, and decreased or absent breath sounds [36,40].

In contrast with CRSCs, whose interpretation is ruled by a threshold, the definitions of normal and abnormal breath sounds on AUS are not clearly stated. It is for this reason that Curtis et al. [36] qualified the interpretation of breath sounds as an art, implying important subjectivity based on the talent and expertise of the operator. Boccardo et al. [40] recently tried to standardize AUS interpretation in pre-weaned dairy calves. They suggested three different definitions: (1) AUSC 1, considering calves positive when there are increased breath sounds, wheezes and crackles, increased bronchial sounds, and pleural friction rubs; (2) AUSC 2, considering calves positive when there are wheezes and crackles, increased bronchial sounds, and pleural friction rubs; and (3) AUSC 3, considering calves positive when there are increased bronchial sounds and pleural friction rubs. The definition with the highest performance was AUSC 2, with a Se of 80% and Sp of 90%. This standardization increased the test’s performance.

In feedlots, an electronic stethoscope (Whisper^®^, https://www.microtechnologies.com/feedyard/whisper-veterinarian-stethoscope-system, 11 January 2024) has been developed to improve objectivity in the interpretation of AUS during BP [41,42,43]. Despite its promising performance in beef cows ≥ 250 kg (Se of 92% [95%BCI = 71; 99] and Sp of 90% [95%BCI = 64; 99]) [42], this tool has not been validated in pre-weaned dairy and veal calves. The results are unpredictable.

### 3.3. Performance and Agreement between Thoracic Auscultation

Table 4 shows the Se and Sp of AUS in pre-weaned dairy and veal calves.

The performance is variable and depends on the study’s definition of a negative or positive test. For example, if we compare the performance of AUS in Buczinski et al. [38] and Buczinski et al. [44], we notice that the addition of “increased breath sounds” such as abnormal breath sounds made AUS more sensitive (from 0.17 to 0.73) while decreasing Sp (from 1 to 0.53) to detect active or inactive BP. 

Interestingly, Boccardo et al. [40] showed that defining AUS as positive when there are wheezes and crackles, increased bronchial sounds, and pleural friction rubs increased AUS’s performance in detecting active BP. With a Se of 82% and a Sp of 91% [40], a total of 20% of sick calves will not be detected in a population with an active BP prevalence of 20%. In the same example, 9% of calves will be declared sick and, therefore, treated with antibiotics without needing them. This performance is superior to that of CRSCs, encouraging the use of AUS with this definition to improve BP diagnosis. 

However, AUS with this definition was performed by only one operator in Boccardo et al. [40]. The agreement between operators was not estimated. We do not know if this promising performance would be the same with other operators. Agreement with AUS was reported in only one study and was poor (indicator of agreement < 0.2). However, the test was not standardized, and interpretation was left to the operators, which could increase the variability [45]. 

Conclusion: Thoracic auscultation is a simple test for veterinarians to detect BP in pre-weaned dairy and veal calves. Recently, the standardization of what is positive or negative on AUS has increased the test’s performance. However, its performance is not perfect, and other diagnostic tests are needed after AUS to detect active BP more accurately. Moreover, agreement needs to be estimated between operators to know the real diagnostic potential of this standardization.

## 4. Imaging Tests

The imaging tests allow for the visualization of lung lesions. Computed tomography (CT) is the most accurate test because it can examine the whole lung without any superimposition. Although it is used in dairy calves, this use is limited because of its cost and because it is only accessible in a hospital setting [46,47,48]. In practice, TUS and TR are used. In comparison with CT, TUS cannot assess deep lesions in the lung parenchyma [49], and anatomical superimposition can occur with TR [50]. That is why both tests are less accurate than CT, but they are less expensive and more accessible.

### 4.1. Thoracic Ultrasonography

#### 4.1.1. Technique of Thoracic Ultrasonography

Thoracic ultrasonography is performed on standing calves. The technique consists of applying a rectal or sectorial probe of 7.5–12 MHz between the ribs and scanning both sides from the 13th intercostal space (ICS) to the 1st or 2nd ICS, from the top to the bottom. To access the right and left cranial parts of the lobes, the probe is inserted behind the thoracic legs [51,52]. Shaving and gel, or spraying 70% isopropyl alcohol or vegetable oil without shaving, can be used to increase contact and improve image quality [31,53,54]. Simplified techniques have been described in order to increase their rapidity when multiple calves need to be assessed by bilateral scanning from the 8th ICS to the 3rd ICS [38]; from the 5th to the 1st right side and the 2nd left side ICS [55]; from the 5th to the 4th ICS, bilaterally [54], or by scanning the entire lung field in one fluent motion [52]. The performance or practicality of these simplified and quick techniques have never been compared. There is, therefore, no standardized technique used by practitioners [56]. 

Globally, TUS is fast (from 4 min for experienced operators [31] to 9 min for less experienced operators [57]); non-invasive, as standing calves are scanned without sedation [31,51]; and costless regarding material because a rectal probe is already accessible to the majority of bovine practitioners [51]. Some authors, such as Ollivett and Buczinski, have questioned the interest in scanning the cranial parts of the lung, especially the left part, stating that it is rarely infected and can be confused with the thymus [51]. The right cranial lobe is often infected [58,59], but its use in a routine exam requires expertise, with its images being more difficult to interpret.

#### 4.1.2. Interpretation of Thoracic Ultrasonography

Some lesions on TUS are frequent but found in both healthy and sick calves, such as comet tails, scar tissue, or B-lines. Other lesions are solely found in sick calves but are infrequent, such as pleural effusion, lung abscesses, or pneumothorax [60]. Lung consolidations are the most present and most specific lesions of BP [38,60] (Figure 2). This lesion is therefore a good indicator of the disease. Additionally, lung consolidations appear very early (2 h) after experimental infection with *Mannheimia haemolytica* and before clinical signs (6 h) [53]. These lesions persist after clinical signs, which can be used to monitor the duration of treatment [61] or detect subclinical calves [53,62,63]. 

Different parameters have been studied to detect BP, including the presence and number of sites with a comet tail, the presence of pleural effusion, the irregularity and/or thickness of the pleura, the depth of lung consolidation corresponding to the thickness in cm of the lesion from the pleura, and the number of lung consolidation sites [38]. Among all of these, the depth of lung consolidation was the simplest and most accurate parameter to detect calves treated for BP [38] or predict a negative outcome in pre-weaned dairy calves [38] and beef cows [64,65]. Additionally, the study of Ollivett et al. [58], performed on a low number of calves (n = 25) without clinical signs, showed a possible correlation between the depth of lung consolidation and viral (<3 cm) or bacterial infections (≥3 cm) [58]. 

Different depth thresholds have been used to interpret lung consolidations, including 0 cm [13,66], 1 cm [30,59], 3 cm [14,67], or 6 cm [68]. These thresholds were chosen subjectively, except by Berman et al. [69], where a threshold of 3 cm for lung consolidation located caudally to the heart was the most specific in detecting active BP [69]. However, this study was performed in a population of low prevalence. It is possible that, in a population of high prevalence, a threshold of 1 cm is better. 

Ollivet and Buczinski described a scoring system according to the extension of the lesions [51]. However, this score has never been developed and validated. Recently, there is an increasing interest in using this scoring system by an infected area (caudal (10th–7th ICS), middle (6th–5th ICS), and cranial (4th–3rd ICS) of both lung sides) [70]. 

#### 4.1.3. Performance and Agreement of Thoracic Ultrasonography

Table 5 shows the Se and Sp of TUS. 

Considering the Se of 0.89 and Sp of 0.95 reported by Berman et al. [69] to detect active BP in a population with 20% BP prevalence, the proportion of calves not diagnosed would be 10%, which is twice and a third less than the false-negative proportion found with CRSCs and AUS, respectively. Additionally, only 5% would be diagnosed by TUS as being sick and unnecessarily treated with antibiotics, which is below the false-positive proportion of CRSCs (from 6% to 14%) and AUS (at 9% with the most accurate definition of Boccardo et al. [40]. 

The interrater agreement of TUS varies according to studies. Two studies showed good agreement (indicators of agreement > 0.6) between different operators of various experiments [57,74]. However, these studies were performed with a small number of operators (n = 3) [57] or rely on image interpretation without considering the challenges in the execution of the technique and capturing of an ideal image to interpret [74]. A more recent field study including 38 operators of various experiments showed fair agreement (indicators of agreement < 0.4) [56]. In this latter study, an improvement in performance was, however, possible with supervised training sessions [56]. 

### 4.2. Thoracic Radiography

Thoracic radiography is an imaging test that allows for lesions in the lung to be visualized. Unlike TUS, TR allows for the whole lung to be visualized, so lesions can be detected deeper in the parenchyma [75,76]. Its major drawback is the superimposition of anatomic structures, making interpretation more difficult, especially in adults [77]. This drawback is, however, less problematic in pre-weaned dairy calves because they are easier to manipulate to realize dorsal or ventral views, or it is easy to put the forelimbs forward and clear the cranial lobes. Currently, the use of TR is often limited to hospital settings despite the recent accessibility of portable machines [48]. The cost of TR is, however, superior to that of TUS. 

#### 4.2.1. Technique of Thoracic Radiography

The first description of the use of the TR technique among dairy calves was performed by Slocombe et al. [78]. In this description, calves were placed under general anesthesia, and all the possible views were performed (lateral (LV), ventro-dorsal (VD), and dorso-ventral (DV)). The technique was then simplified to increase its practicality in the field. Firstly, portable machines are used in the field with the following setup: intensity of 150 mAs, beam energy of 80 kVp, exposure time of 0.13 ms, tape of 35 × 43 cm placed 100 cm from the cathode, grid of 3:1 [79,80]. Secondly, only the LV is commonly performed on standing calves without sedation [48,81]. To limit superimposition, the forelegs are extended forward to clear the cranial lobes [81,82]. Recently, Shimbo et al. [81] described a rapid technique that can be performed by a single person. In this technique, the calf is restrained in a chute and a unilateral forelimb is pulled cranially with the contralateral forelimb tied to the chute; the forelimbs are then spread craniocaudally, as in a scissor position. The tape is fixed on the chute, allowing for this technique to be applied to several calves. However, this technique needs to be used on calm calves, which are easy to manipulate and can tolerate the position during the radiography. This technique could, indeed, be difficult on more agitated calves (young age, absence of dullness) or in calves in respiratory distress. 

#### 4.2.2. Interpretation of Thoracic Radiography

Several patterns could be visualized on TR during BP: bronchial pattern, when there are prominent and/or thickened bronchial walls; interstitial pattern, when there is a diffuse increased opacity of the lung parenchyma causing loss of definition of the vascular structures; and alveolar pattern, when the lung presents a soft tissue opacity that completely obscures the identification of the vascular structures, borders of the heart, or diaphragm; and nodular pattern, when there are one or more soft tissue opacity structures measuring up to 3 cm in diameter. Pleural effusion and pneumothorax are also easily visualized on TR [48,81,82,83]. Among those lesions, the alveolar pattern is the most present lesion during BP (Figure 3) [48,79,80,81].

The alveolar pattern can, however, be confused with the superimposed anatomic structures, especially because of the forelegs when the calf is standing [81,82]. This could lead to a false positive diagnosis (calves do not actually have any lesions), impacting the Sp of the test [83]. Extending the forelegs forward or adding DV or VD views can limit those superimposition issues [78,81,82]. Unlike TUS, the interpretation of BP lesions on TR is not simple for practitioners. Their detection and interpretation need experts [81]. To universalize its use by practitioners in the field, a simplification of its interpretation is needed.

#### 4.2.3. Performances and Agreement of Thoracic Radiography

Table 6 shows the Se and Sp of TR. The performance of TR is similar to that of TUS. Moreover, comparisons of TR and TUS show no difference between both tests in detecting lung lesions or active BPI [72,73]. Moreover, agreement was only estimated between experts and was fair [81]. This performance is, therefore, likely to be variable if no experimental operator performs and interprets TR. For all these reasons, TUS seems more appropriate in the detection of BP in the field.

Conclusion: The performance of imaging tests is superior to that of CRSCs and AUS for detecting BP. Thoracic ultrasonography is easier and more appropriate for bovine practitioners. However, more studies are needed to know if a lung lesion is from a viral or bacterial infection and if antibiotics are needed. 

## 5. Conclusions

This study reviewed the techniques of CRSC, AUS, TUS and TR; their performance (Se, Sp, agreement between operators); and their limits of estimation (biases, precision, status investigated) to detect BP in pre-weaned dairy and veal calves. We notice that detecting BRD with CRSCs and AUS can lead to diagnosis errors. The use of imaging tests, especially TUS, either after those tests or alone, can improve BP detection. The standardization of those tests with supervised field-training supervised sessions could improve their use in the field and their performance. Future investigations are therefore needed to continue to improve those tests, such as individually accurate CRSC, a repeatable definition of AUS, and distinction between viral vs. bacterial lung lesions on TUS. 

## Figures and Tables

**Figure 1 animals-14-00329-f001:**
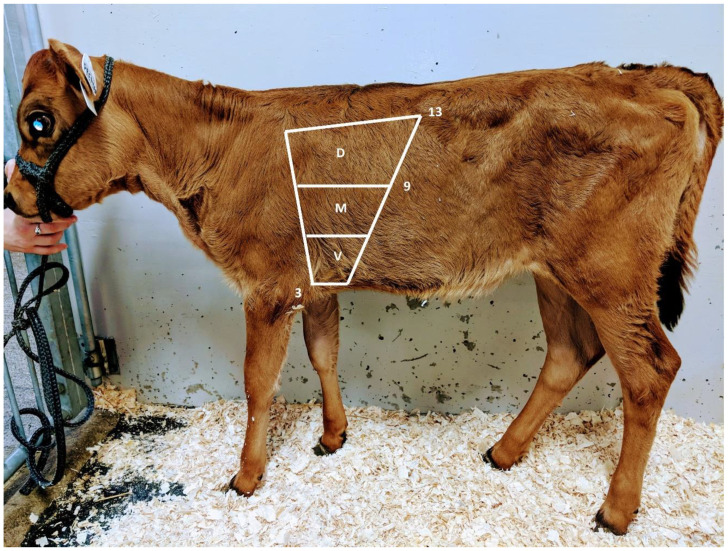
Illustration of thoracic auscultation sites. The total auscultation area ranges from the 13th rib in the dorso-caudal region through the middle of the 9th rib to approximately the 3rd intercostal space, located below the axillary region of the cranial thorax. The total area can be divided into dorsal (D), median (M) to ventral (V) thirds. The auscultation area was identical for both the right and left hemithorax.

**Figure 2 animals-14-00329-f002:**
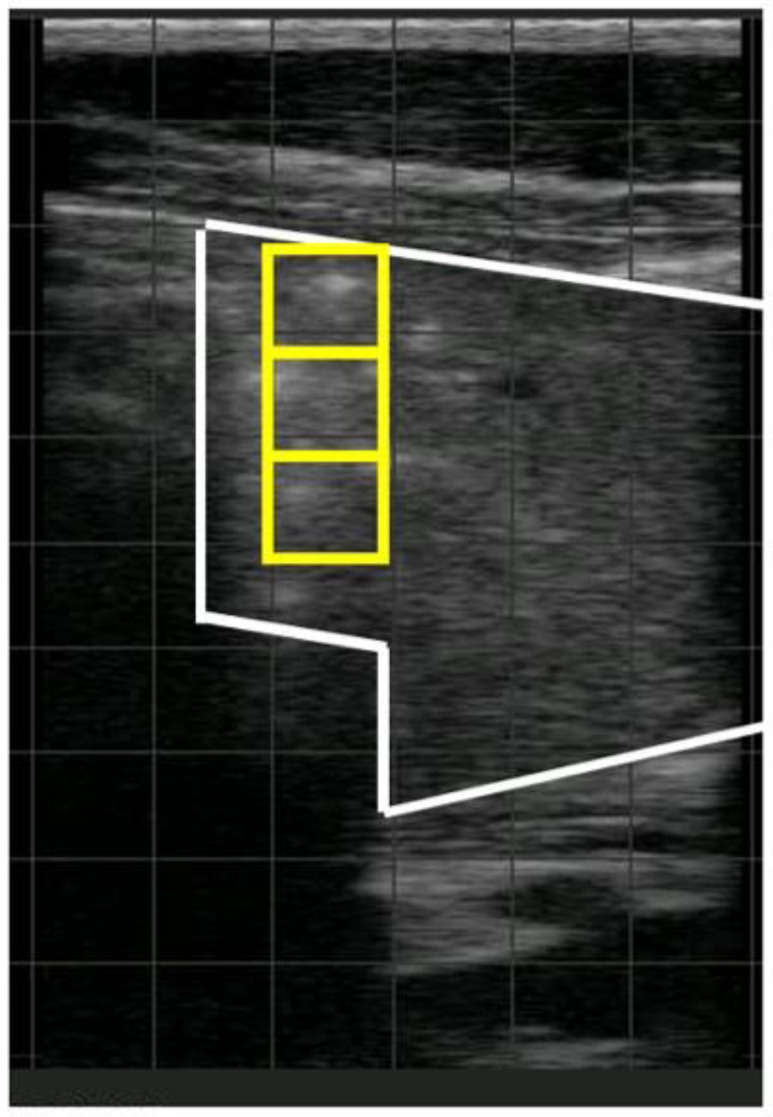
Image of lung consolidation on thoracic ultrasonography (white frame). The air in the alveoli has been replaced by inflammatory, tumoral, or cicatricial material. Hypoechoic tissue, like hepatic parenchyma on ultrasound, is present in lung parenchyma. This lesion is present during infectious bronchopneumonia, pneumonia by aspiration, and, less frequently, during lung contusion or lung metastases. The depth of lung consolidation corresponds to the thickness in cm of the lesion from the pleura. In this image, the depth is superior to 3 cm (yellow squares).

**Figure 3 animals-14-00329-f003:**
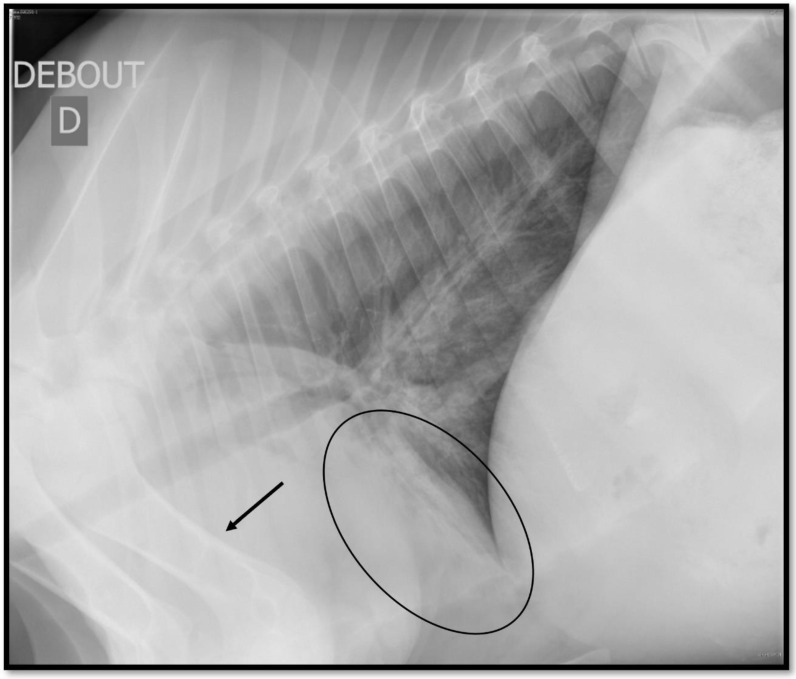
Image of an alveolar pattern on thoracic ultrasonography (circle). The alveoli are filled by an inflammatory material, either tumoral or cicatricial. An area of soft-tissue opacity is present, hiding the pulmonary vessels and the structures in the mediastinum like the cardiac borders. One of the drawbacks of thoracic radiography is the anatomical superimposition, which can hide or be confused with an alveolar pattern like the superimposition of the forelimbs (arrow).

**Table 1 animals-14-00329-t001:** Principal clinical respiratory scoring systems used in pre-weaned dairy and veal calves to detect bovine respiratory disease.

Clinical Respiratory Scoring System	Developing Population	Predictors	Assessment of Predictors	Weight ^1^	Interpretation of the Final Score
WISCONSIN SCORE McGuirk [26]	Pre-weaned dairy calves housed in group.	5 predictors Rectal temperature Nasal discharge Ocular discharge Cough Ear droop or head tilt	Severity (0–3) ^2^	♦ x 1 for all predictors	Subjective threshold. two thresholds reported: ≥6 [25] ≥5 [26]
CALIFORNIA SCORE Love et al. [27]	Pre-weaned dairy calves housed individually.	6 predictors The same 5 predictors as the Wisconsin score + Abnormal breathing	Presence or absence	♦ x 2 Ocular discharge, cough, abnormal breathing, rectal temperature (≥39.2 °C) ♦ x 4 Nasal discharge ♦ x 5 Ear droop or head tilt	Threshold statistically estimated to optimize the performance of the score. Threshold ≥ 5
CALIFORNIA SCORE from QUÉBEC Buczinski et al. [28]	Pre-weaned dairy calves housed individually or in group.	6 predictors The same 6 predictors as the California score	Presence or absence	♦ x 20 Abnormal breathing ♦ x 16 Ear droop or head tilt, cough ♦ x 10 Nasal discharge ♦ x 7 Rectal temperature ♦ x − 1 Ocular discharge	Threshold statistically estimated to optimize the performance of the score. Thresholds vary according to the prevalence in a group. Threshold varying from 9 to 16. The more the prevalence is elevated, the more the threshold is decreased.
BERMAN SCORE–VEAL CALVES Berman et al. [29]	Veal calves housed individually.	3 Predictors Cough Rectal temperature Ear droop or head tilt	Presence or absence	♦ x 10 Cough ♦ x 9 Ear droop or head tilt ♦ x 6 Rectal temperature (≥39.7 °C)	Threshold statistically estimated to reduce economic losses due to a wrong diagnosis. Individual: ≥15 or the presence of 2 signs on 3. Group: 3 positive calves among 10 sampled calves.

^1^ Into this column, each weight is indicated by the multiplying symbol (x). The final score is obtained by multiply the weight by 1 if the predictor is present, 0 if the predictor is absent, and adding each value. ^2^ Rectal temperature: 0 = 37.8–38.2 (100–100.9), 1 = 38.3–38.8 (101–101.9), 2 = 38.9–39.3 (102–102.9), 3 = ≥39.4 (>103.0); Nasal discharge: 0 = serous, 1 = small amount of unilateral cloudy, 2 = bilateral cloudy or excessive mucus, 3 = copious bilateral mucopurulent; Cough: 0 = none, 1 = induce single, 2 = induce repeated or occasional spontaneous cough, 3 = repeated spontaneous coughing; Eye or ear: 0 = normal, 1 = mild discharge, 2 = bilateral purulent discharge or unilateral ear drop, 3 = head tilt or both ears dropped. This is the traditional and simplest Wisconsin score. Noted that another Wisconsin score, which is more extensive and applicable to assess calves’ health, is available, including clinical signs such as diarrhea or umbilicus.

**Table 2 animals-14-00329-t002:** Sensitivities (Se) and specificities (Sp) of clinical respiratory scoring systems used in pre-weaned dairy and veal calves to detect bovine respiratory disease.

Clinical Respiratory Scoring System	Reference	Study Population	Comparator Test to Confirm the Disease	Performance	Limits
WISCONSIN SCORE	Buczinski et al. [30]	Pre-weaned dairy calves housed in group or individually from a population of high prevalence [30] or low prevalence [31].	Latent class Bayesian analysis using imperfect thoracic ultrasonography results (positive if lung consolidation ≥ 1 cm).	Se = 0.62 (95%BCI: 0.48; 0.76) Sp = 0.74 (95%BCI: 0.65; 0.83)	♦ Imprecisions around Se and Sp values. ♦ Performance for BRD (upper infection, active and inactive BP).
	Love et al. [32]	Pre-weaned dairy calves housed individually, with calves randomly selected for the estimation of screening sensitivity (SSe), and suspected sick calves selected for an estimation of confirmatory sensitivity (CSe).	Composite standard test = 2 imperfect tests interpreted in parallel. ♦ Thoracic ultrasonography (positive if extensive or focal consolidation, abscess > 2 cm, pleural effusion). ♦ Thoracic auscultation (positive if abnormal breath sounds are present).	SSe = 0.46 (95%CI: 0.39; 0.53) CSe = 0.71 (95%CI: 0.64; 0.78) Sp = 0.91 (95%CI: 0.87; 0.94)	♦ Spectral bias–case-control design. ♦ Classification bias–imperfect comparator test. ♦ Performance for BRD (upper infection, active and inactive BP).
	Lowie et al. [33]	Database from 297 dairy calves, including pre-weaned and weaned calves, and 399 veal calves.	Single comparator test: thoracic ultrasonography (positive if consolidation ≥ 1 and ≥3 cm).	Pre-weaned dairy and veal calves Se = 0.27 (≥1 cm) and 0.22 (≥3 cm) Sp = 0.94 (≥1 cm) and 0.86 (≥3 cm)	♦ No CI calculation to assess precision of values. ♦ Classification bias–imperfect comparator test. ♦ Performance for BRD (upper infection, active and inactive BP).
CALIFORNIA SCORE	Love et al. [32]	Pre-weaned dairy calves housed individually, with calves randomly selected for the estimation of screening sensitivity (SSe), and suspected sick calves selected for an estimation of confirmatory sensitivity (CSe).	Composite standard test = 2 imperfect tests interpreted in parallel. ♦ Thoracic ultrasonography (positive if extensive or focal consolidation, abscess > 2 cm, pleural effusion). ♦ Thoracic auscultation (positive if presence of abnormal sounds).	SSe = 0.47 (95%CI: 0.40; 0.54) CSe = 0.73 (95%CI: 0.65; 0.79) Sp = 0.87 (95%CI: 0.83; 0.91)	♦ Spectral bias–case-control design. ♦ Classification bias–imperfect comparator test. ♦ Performance for BRD (upper infection, active and inactive BP).
	Lowie et al. [33]	Database from 297 dairy calves, including pre-weaned and weaned calves, and 399 veal calves.	Single comparator test: thoracic ultrasonography (positive if consolidation ≥ 1 and ≥3 cm).	Pre-weaned dairy and veal calves Se = 0.33 (≥1 cm) and 0.23 (≥3 cm) Sp = 0.83 (≥1 cm) and 0.86 (≥3 cm)	♦ No CI calculation to assess the precision of values. ♦ Classification bias–imperfect comparator test. ♦ Performance for BRD (upper infection, active and inactive BP).
CALIFORNIA SCORE from QUÉBEC	Buczinski et al. [28]	Pre-weaned dairy calves housed individually or in group.	Latent class Bayesian analysis using imperfect results of thoracic ultrasonography (positive if lung consolidation ≥ 1 cm)	Se from 0.83 to 0.67 for a threshold from 9 to 13, respectively, according to the prevalence in the group. Sp from 0.69 to 0.83 for a threshold from 9 to 13, respectively, according to the prevalence in the group.	♦ Performance for BRD (upper infection, active and inactive BP).
BERMAN SCORE–VEAL CALVES	Berman et al. [29]	Veal calves housed individually.	Latent class Bayesian analysis using imperfect results of thoracic ultrasonography (positive if lung consolidation ≥ 3 cm) and haptoglobin dosage (positive if ≥0.25 mg/L).	Se = 0.31 (95%BCI: 0.10; 0.70) Sp = 1.00 (95%BCI: 0.99; 1.0)	♦ Imprecisions around Se values because of the low prevalence. ♦ Performance of active BP.

**Table 3 animals-14-00329-t003:** Sounds audible during the thoracic auscultation of pre-weaned dairy and veal calves.

Audible Sound	Definition	Pathologic Signification
Normal breath sounds	Normal velocity of respiratory gases.	Physiologic
Increased breath sounds	Increase of the velocity of respiratory gases secondary to the respiratory frequency increased, or the breathing depth increased.	♦ Physiologic (e.g., intense training, excitation, elevated environmental temperature). ♦ Pathologic (e.g., fever, acidosis, beginning of pulmonary oedema).
Increased bronchial sounds	Sounds audible in the pulmonary area, such as tracheal sounds. Increased bronchial sounds occur when bronchi are surrounded by consolidated lung parenchyma, enhancing sound amplification.	Pathologic (infectious bronchopneumonia with lesions of lung consolidation with bronchograms (i.e., bronchi are not filled by liquid)).
Crackles	Clicking, rattling, or cracking noises. Two phenomena can cause this sound: (1) liquid bubble bursting into respiratory tracts, or (2) respiratory tract that remains closed during a part of the inspiration and opens suddenly.	Pathologic (pulmonary edema, interstitial pneumonia).
Wheezes	Vibration of respiration of the airways or air passing through the narrowed airways.	Pathologic (allergic pneumonia, infectious bronchopneumonia, interstitial pneumonia).
Pleural friction rubs	Noises similar to the use of sandpaper are caused by inflammation of the pleural cavity and the friction between both visceral and parietal pleura.	Pathologic (pleurobronchopneumonia).
Decreased breath sounds	Decrease in the velocity of respiratory gases caused by: (1) obstruction of the transmission of breath sounds (fluid accumulation (pleural effusion), air (pneumothorax), pus (pleural abscess or pyothorax), obesity); (2) severe lung consolidation with bronchograms (i.e., bronchi filled of liquids); (3) superficial respiration (pain, weakness, meningoencephalitis).	♦ Physiologic (obesity). ♦ Pathologic (infectious bronchopneumonia, pleurobronchopneumonia, pain, weakness, meningoencephalitis).
Sounds from thoracic area	Respiratory grunts, cardiac sounds, ruminal sounds, cutaneous friction sounds.	

**Table 4 animals-14-00329-t004:** Sensitivities (Se) and specificities (Sp) of the thoracic auscultation to detect infectious bronchopneumonia (BP) in pre-weaned dairy and veal calves.

Study	Population	Definition of a Positive Test	Comparator Test to Confirm the Disease	Performance	Limits
Buczinski et al. [38]	Pre-weaned dairy calves	Wheezes, crackles, decreased breath sounds, pleural friction rubs.	Thoracic ultrasonography (positive if lung consolidation ≥ 1 cm) used as gold standard test.	Se = 0.06 between 0 to 0.17 according to auscultation area Sp = 0.99 between 0.97 to 1.0 according to auscultation area	♦ Classification bias: imperfect comparator test. ♦ Performance for active and inactive BP.
Buczinski et al. [44]	Veal calves around three weeks old	Wheezes, crackles, decreased breath sounds, pleural friction rubs, and increased breath sounds.	Latent class Bayesian analysis using imperfect results of thoracic ultrasonography (positive if lung consolidation ≥ 1 cm).	Se = 0.73 (95%BCI: 0.50; 0.96) Sp = 0.53 (95%BCI: 0.43; 0.64)	♦ Imprecisions around Se and Sp values. ♦ Performance for active and inactive BP.
Pardon et al. [45]	3 sections with 8–10 veal calves of 5, 10, and 13 weeks old.	According to the operator.	Thoracic ultrasonography (positive if lung consolidation ≥ 1 cm) used as gold standard test.	Se = 0.63 (SD = 0.2) Sp = 0.46 (SD = 0.3) SD: Standard deviation	♦ Classification bias: imperfect comparator test. ♦ Imprecisions around Se and Sp values. ♦ Performance for active and inactive BP.
Boccardo et al. [40]	330 pre-weaned and weaned dairy calves	AUSC 1: Increased breath sounds, wheezes, crackles, increased bronchial sounds, and pleural friction rubs.	Latent class Bayesian analysis using imperfect results of thoracic ultrasonography (positive if lung consolidation ≥ 3 cm) and Wisconsin Score (positive if score ≥ 5).	AUSC 1 Se = 0.90 (95%BCI: 0.81; 0.98) Sp = 0.57 (95%BCI: 0.47; 0.72)	♦ Imprecisions around Se and Sp values ♦ Performance for active and inactive BP
		AUSC 2: Wheezes, crackles, increased bronchial sounds, and pleural friction rubs.		AUSC 2 Se = 0.82 (95%BCI: 0.69; 0.95) Sp = 0.91 (95%BCI: 0.81; 0.99)	
		AUSC 3: Increased branchial sounds and pleural friction rubs.		AUSC 3 Se = 0.68 (95%BCI: 0.56; 0.83) Sp = 0.99 (95%BCI: 0.94; 1.0)	

**Table 5 animals-14-00329-t005:** Sensitivities (Se) and specificities (Sp) of thoracic ultrasonography to detect infectious bronchopneumonia (BP) in pre-weaned dairy and veal calves.

Study	Population	Definition of a Positive Test	Comparator Test to Confirm the Disease	Performance	Limits
Rabeling et al. [66]	Weaned dairy calves with severe BP.	Comet tails, abscesses, consolidation (>0 cm).	Test gold standard = necropsy	Se = 0.85 Sp = 0.98	♦ Spectral bias, case-control design and cases being severely or chronically infected.
Jung and Bostedt. [71]	Pre-weaned dairy calves of 0 to 14 days old.	Comet tails, abscesses, pleural effusion, pneumothorax, consolidation (>0 cm).	Imperfect single test = thoracic radiography	Se = 0.77 Sp = 1.0	♦ Classification bias: imperfect comparator test. ♦ Spectral bias–case-control design. ♦ Performance for active BP.
Ollivett et al. [58]	Pre-weaned dairy calves without clinical signs (score of Wisconsin < 5).	Comet tails, abscesses, pleural effusion, pneumothorax, consolidation (>0 cm).	Test gold standard = necropsy	Se = 0.94 (95%CI: 0.69; 1) Sp = 1.0 (95%CI: 0.64; 1)	♦ Spectral biases, including solely subclinical cases. ♦ Imprecision around Se and Sp values. ♦ Performance for inactive and active BP.
Buczinski et al. [30]	Pre-weaned dairy calves housed in group or individually from a population of high prevalence [30] or low prevalence [31].	Consolidation (≥1 cm).	Latent class Bayesian analysis performance when estimating the performance of Wisconsin score and using priors from experts regarding performance of thoracic ultrasonography.	Se = 0.79 (95%BCI: 0.66; 0.91) Sp = 0.94 (95%BCI: 0.88; 0.98)	♦ Previous performances included thoracic ultrasonography and a small number of calves (n = 191), which could have influenced the results of Bayesian analysis. ♦ Imprecision around Se values (low prevalence). ♦ Performance for inactive and active BP.
Buczinski et al. [44]	Veal calves around three-weeks old.	Consolidation (≥1 cm).	Latent class Bayesian analysis performance to estimate the performance of thoracic auscultation and using priors from experts for the performance of thoracic ultrasonography.	Se = 0.77 (95%BCI: 0.60; 0.89) Sp = 0.93 (95%BCI: 0.87; 0.97)	♦ Previous performances used thoracic ultrasonography and a small number of calves (n = 209), which could have influenced the results of Bayesian analysis. ♦ Imprecision around Se values (low prevalence). ♦ Performance for inactive and active BP.
Berman et al. [69]	Veal calves around three-weeks old and pre-weaned dairy calves.	Different thresholds of consolidation (0, 1 or 3 cm) et different sites (caudal versus cranial).	Latent class Bayesian analysis performance to estimate performance of thoracic ultrasonography and using priors from experts for performance of Wisconsin scoer and haptoglobin dosage.	Best performances for the threshold of 3 cm, caudal Se = 0.89 (95% BCI: 0.55; 1.0) Sp = 0.95 (95% BCI: 0.92; 0.98)	♦ Imprecision around Se values (low prevalence). ♦ Performance for active BP.
Berman et al. [72]	Dairy calves weighting ≤ 100 kg.	All lesions, consolidation (≥1 cm).	Diagnostic panel including 3 experts in bovine medicine with an inter-operator agreement of 0.58.	Se = 0.84 (95%CI: 0.60; 0.97) Sp = 0.74 (95%CI: 0.57; 0.86)	♦ Imprecision around Se and Sp values (low number of calves (n = 50) and lack of power). ♦ Performance for active BP.
Berman et al. [73]	Dairy calves weighting ≤ 100 kg.	All lesions, consolidation (≥1 cm).	Latent class Bayesian analysis in two-steps using CT scan as gold-standard test on positive calves on thoracic ultrasonography and/or thoracic radiography.	Se = 0.81 (95%BCI: 0.65; 0.92) Sp = 0.90 (95% BIC: 0.81; 0.96)	♦ Imprecision around Se and Sp values (low number of calves (n = 50) and lack of power). ♦ Performance for active and inactive BP (lung lesions).

**Table 6 animals-14-00329-t006:** Sensitivities (Se) and specificities (Sp) of thoracic radiography to detect infectious bronchopneumonia (BP) in pre-weaned dairy and veal calves.

Study	Population	Definition of a Positive Test	Comparator Test to Confirm the Disease	Performances	Limits
Berman et al. [72]	Dairy calves weighting ≤ 100 kg.	All the lesions	Diagnostic panel including 3 experts in bovine medicine with an inter-operator agreement of 0.58.	Se = 0.89 (95%CI: 0.67; 0.99) Sp = 0.58 (95%CI: 0.39; 0.75)	♦ Imprecision around Se and Sp values (low number of calves (n = 50) and lack of power). ♦ Performance for active BP.
Berman et al. [73]	Dairy calves weighting ≤ 100 kg.	All the lesions	Latent class Bayesian analysis in two-steps using CT scan as gold-standard test on positive calves on thoracic ultrasonography and\or thoracic radiography.	Se = 0.86 (95%BCI: 0.62; 0.99) Sp = 0.89 (95%BCI: 0.67; 0.99)	♦ Imprecision around Se and Sp values (low number of calves (n = 50) and lack of power). ♦ Performance for active and inactive BP (lung lesions).

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
