# Peer review of "Literature Review of the Principal Diagnostic Tests to Detect Bovine Respiratory Disease in Pre-Weaned Dairy and Veal Calves"

_animals, 2024, doi:10.3390/ani14020329_

Round 1

Reviewer 1 Report

Comments and Suggestions for Authors

Dear editor, dear author(s)

Thank you to give me the opportunity to review this work. The present article is well-written and provides a summary of the existing detection methods to identify BRD, including clinical scorecards, auscultation, radiography and ultrasound. Although this review may contribute to the body of existing literature, there are same major remarks that need to be addressed.  

Best wishes.

Major remarks:

This is narrative/systematic review, that should be explicitly clarified in the methodology section. According the type of review, there exist guidelines for conducting reviews, and it seems that adherence to these guidelines may not have been consistently followed. For example there is no clear specification regarding the article search strategy.

While the review takes a narrative/systematic approach, there may be an opportunity to explore meta-analysis possibilities, particularly concerning the sensitivity and specificity of ultrasound etc. (forest plot could be used). In comparison with existing work, by Buczinski and Pardon, reveals that the added papers predominantly consist of the author's own contributions. It is crucial to verify that no other relevant work has been missed. For example, there is  an alternative ultrasound technique published (Pardon et al. 2019, Jourquin et al 2022). Furthermore, inter-observer agreement in ultrasound, an abstract from the recent WBC is noteworthy. It is recommended to extend the search to include abstracts from main cattle conferences for a more comprehensive coverage of the topic.

The reader may find the interchangeability of the terms BRD and BP confusing. Clarifying these terms is essential for better comprehension of the article. Bovine Respiratory Disease (BRD) is a broad and historical term (Buczinski and Pardon 2020) that includes both upper respiratory tract infections and lower respiratory tract infections, with bronchopneumonia serving as a proxy for the latter in this study. Further, clinical score cards (Wisconsin, California) are developed for the detection of BRD and not specifically for the detection of BP. Providing this clarification should enhance the reader's understanding of the content. (In extension of line 52-56)

Specific comments:

Line 32-34: A more recent study about antimicrobial use in the field for BRD has been published (Lowie et al. 2022)

Line 64: The observer is not the sole factor to consider in the interpretation of clinical signs, another important aspect to consider is the variability in the expression of clinicals signs used for the detection of BRD within and between calves. 

Table 1: It would help the reader to understand the Wisconsin score better to add the classification of the assessment of predictors (e.g. nasal discharge: normal serous discharge, small amount of unilateral cloudy discharge, etc.).

Table 1: There is as well a Wisconsin score available that added fecal score.

Line 111-113: This extrapolation should be approached with more nuance, as the Wisconsin classification is more susceptible to subjectivity compared to a dichotomous system like the California score.

Line 114-115: Again, this statement should be approached with more nuance

Line 118-120: The CRSC systems were initially designed for the detection of BRD. Including this information would enhance the completeness of the conclusion.

Line: 139-140: Is there a reference supporting this statement?

Line 156: Perhaps it is not categorized as a “sound”, however the absence of sound may fit into this story.

Line 222-223: Not all techniques for rapid thoracic ultrasonography are discussed in the manuscript. It would be beneficial to the manuscript to include all available techniques. For example, quick thoracic ultrasonography (qTUS) is as a technique used both in practice and in literature (Pardon et al. 2019, Jourquin et al 2022). Is there anything known about the performance of the different techniques?

Line 238: Other lung ultrasound "lesions" to take into account are lung abscesses and scar tissue.

Line 255: There is as well an increasing interest in the use of area infected. (Fiore et al. 2022)

Line 281: Nuance statement, because the studies cited either have a small sample size or rely on image interpretation, with a notable challenge being the execution of the technique and capturing an ideal image to interpret. The agreement between observers improved significantly with proper training, particularly through supervised practical sessions. (Pardon et al 2022, WBC)

Line 294: Extra limitation to consider is the economic aspect, given that the cattle industry is heavily influenced by economic factors.

Line 320: Elaborate a bit more on the different findings in TR.

Line 394: For the implementation of TUS in the field training and standardization is key

Author Response

We thank both reviewers for their relevant comments. Please find included below specific explanations addressing each comment or suggestion made by both reviewers. 

Reviewer 1

Major remarks:

Comment 1: This is narrative/systematic review, that should be explicitly clarified in the methodology section. According to the type of review, there exist guidelines for conducting reviews, and it seems that adherence to these guidelines may not have been consistently followed. For example, there is no clear specification regarding the article search strategy.

Response 1: I thank the reviewer for this relevant comment. Indeed, the type of reviews was not mentioned in the title or elsewhere. This manuscript is a literature review according to the following definition of Grant and al.:

“a literature review involves some process for identifying materials for potential inclusion— whether or not requiring a formal literature search—for selecting included materials, for synthesizing them in textual, tabular or graphical form and for making some analysis of their contribution or value.”

According to the definition, details of how references were found are not obligatory. I added “literature review” in the title, as recommended.

Grant MJ, Booth A. A typology of reviews: an analysis of 14 review types and associated methodologies. Health information & libraries journal. 2009 Jun;26(2):91-108.

Comment 2: While the review takes a narrative/systematic approach, there may be an opportunity to explore meta-analysis possibilities, particularly concerning the sensitivity and specificity of ultrasound etc. (forest plot could be used).

Response 2:  Thank you for this excellent suggestion. In this study, I wanted to review the principal diagnostic test to detect BRD in pre-weaned calves. We can notice that data about Se and Sp are variable according to the tests, with many studies for CRSC and TUS, and only a few studies for AUS and TR. It seems that meta-analysis could be done only for CRSC and TUS. I, therefore, did not perform this type of analysis to describe each test equally. It is, however, a very good suggestion for future studies about Se and Sp of CRSC or TUS, specifically.

Comment 3:  In comparison with existing work, by Buczinski and Pardon, reveals that the added papers predominantly consist of the author's own contributions. It is crucial to verify that no other relevant work has been missed. For example, there is an alternative ultrasound technique published (Pardon et al. 2019, Jourquin et al 2022). Furthermore, inter-observer agreement in ultrasound, an abstract from the recent WBC is noteworthy. It is recommended to extend the search to include abstracts from main cattle conferences for a more comprehensive coverage of the topic.

Comment 3: Thank you for this comment. Actually, both references mentioned by the reviewer were already included in the manuscript (number 51 for Pardon et al. 2019, and number 59 for Jourquin et al., 2022). More specifically, the reference of Pardon et al. 2019 was cited to describe the technique to scan left lobes on TUS, lines 226-227. The reference of Jourquin et al. 2022 was cited to describe the tool used to monitor treatment response on TUS (lung reaeration). I added those references to describe alternative ultrasound techniques in the revised manuscript, as recommended, lines 233-234.

I thank the reviewer for this remark about inter-agreement in ultrasound. I was aware of these interesting results (that are being revised for publication in Vet Records).  I was just waiting for the official publication but the suggestion of citing WBC abstract conference is very good. I added this point and this reference in the text, lines 234-236 and 295-302.

Comment 4: The reader may find the interchangeability of the terms BRD and BP confusing. Clarifying these terms is essential for better comprehension of the article. Bovine Respiratory Disease (BRD) is a broad and historical term (Buczinski and Pardon 2020) that includes both upper respiratory tract infections and lower respiratory tract infections, with bronchopneumonia serving as a proxy for the latter in this study. Further, clinical score cards (Wisconsin, California) are developed for the detection of BRD and not specifically for the detection of BP. Providing this clarification should enhance the reader's understanding of the content. (In extension of line 52-56)

Response 4: I thank the reviewer for this relevant comment. I clarified the definitions in the text, lines 22-26. I used the term BRD to talk about infection of both the upper and lower respiratory tract, and BP to talk about lower respiratory tract infection. I also used the term BRD to describe the performance of the clinical scores, as recommended.

Specific comments:

Comment 5: Line 32-34: A more recent study about antimicrobial use in the field for BRD has been published (Lowie et al. 2022)

Response 5: I thank the reviewer for suggesting this interesting reference. I read attentively this study and I did not find any information on whether the use of antimicrobials was for outbreaks in calves, adult cows, or both (whole herd). Moreover, this study is for outbreaks specifically. Because my study includes pre-weaned calves specifically, I do not think that this study is appropriate to cite.   

Comment 6: Line 64: The observer is not the sole factor to consider in the interpretation of clinical signs, another important aspect to consider is the variability in the expression of clinicals signs used for the detection of BRD within and between calves.

Response 6: Thank you for this relevant comment. I agree with the fact that the observer is not the sole factor to consider in the observation and interpretation of clinical signs. I reworded the sentence to clarify that point in the text, lines 67-68.

Comment 7: Table 1: It would help the reader to understand the Wisconsin score better to add the classification of the assessment of predictors (e.g., nasal discharge: normal serous discharge, small amount of unilateral cloudy discharge, etc.).

Response 7: We detailed the assessment of predictors for the Wisconsin score in a footnote in Table 1, as recommended.

Comment 8: Table 1: There is as well a Wisconsin score available that added fecal score.

Response 8: I perfectly agree. However, this article is about respiratory disease and not calf health assessment. That is why I restrained the Wisconsin score to BRD Clinical signs. I, however, added a footnote to inform the readers about this point.

Comment 9: Line 111-113: This extrapolation should be approached with more nuance, as the Wisconsin classification is more susceptible to subjectivity compared to a dichotomous system like the California score.

Response 9: I thank the reviewer for this remark. I decided to remove the sentence, lines 122-123, given that the comparison is different.

Comment 10: Line 114-115: Again, this statement should be approached with more nuance.

Response 10: I reworded the sentence, lines 121-123.

Comment 11: Line 118-120: The CRSC systems were initially designed for the detection of BRD. Including this information would enhance the completeness of the conclusion.

Response 11: As I responded in comment 4, I specified "BRD" to talk about the performance of CRSC throughout the manuscript, as recommended.

Comment 12:  Line: 139-140: Is there a reference supporting this statement?

Response 12: Thank you for this point. I quote the reference of Buczinski et al. 2014, line 149.

Buczinski S, Forté G, Francoz D, Bélanger AM. Comparison of thoracic auscultation, clinical score, and ultrasonography as indicators of bovine respiratory disease in preweaned dairy calves. Journal of veterinary internal medicine. 2014 Jan;28(1):234-42.

Comment 13: Line 156: Perhaps it is not categorized as a “sound”, however the absence of sound may fit into this story.

Response 13: Thank you for this comment. I added this information in the text, line 165.

Comment 14: Line 222-223: Not all techniques for rapid thoracic ultrasonography are discussed in the manuscript. It would be beneficial to the manuscript to include all available techniques. For example, quick thoracic ultrasonography (qTUS) is as a technique used both in practice and in literature (Pardon et al. 2019, Jourquin et al 2022). Is there anything known about the performance of the different techniques?

Response 14: Thank you for this comment. I added a quick description of the technique of Pardon et al. 2019, line 233. I did not mention Jourquin et al 2022, because the authors used the technique of Pardon et al. 2019. To the best of my knowledge, no comparison was done between those different quick TUS techniques. There is, therefore, no standardization of the technique between practitioners. I added a sentence about this relevant point, lines 234-235.  

Comment 15: Line 238: Other lung ultrasound "lesions" to take into account are lung abscesses and scar tissue.

Response 15: I added those two types of lesions, lines 247-248.

Comment 16: Line 255: There is as well an increasing interest in the use of area infected. (Fiore et al. 2022)

Response 16: I thank the reviewer for suggesting this interesting reference. I added a sentence about this technique, lines 279-281.

Comment 17: Line 281: Nuance statement, because the studies cited either have a small sample size or rely on image interpretation, with a notable challenge being the execution of the technique and capturing an ideal image to interpret. The agreement between observers improved significantly with proper training, particularly through supervised practical sessions. (Pardon et al 2022, WBC)

Response 17: I thank the reviewer for this relevant point. I modified the sentence to bring those nuances in the inter-rater agreement results and I cited the recommended reference, lines 294-301.

Comment 18: Line 294: Extra limitation to consider is the economic aspect, given that the cattle industry is heavily influenced by economic factors.

Response 18: I added this point, line 310.

Comment 19: Line 320: Elaborate a bit more on the different findings in TR.

Response 19: I added the definitions of the lesions finding on TR during BP, lines 329-336.

Comment 20: Line 394: For the implementation of TUS in the field training and standardization is key.

Response 20: I totally agree with this comment not only for TUS but also for AUS and for CRSC. I added this point in the conclusion, lines 370-372.

Reviewer 2 Report

Comments and Suggestions for Authors

The manuscript submitted for evaluation is an interesting guide to the most important techniques used in the diagnosis of infectious bronchopneumonia in calves.

Minor comments:
Line 12, 77, table 1, the authors should distinguish BP from BRD throughout the manuscript.

Line 38 ,techniques’ will be more adequate than ,tests'.

Figure 1 there is no definition for '3'.

Line 258 please correct ,the’ with a lowercase letter.

Table 5, please correct 'consolidation' with a lowercase letter.

Author Response

Reviewer 2

Minor comments:

Comment 1: Line 12, 77, table 1, the authors should distinguish BP from BRD throughout the manuscript.

Response 1: I thank the reviewer for this relevant comment. I clarified the definitions of BRD and BP in the text, lines 22-26. I used the term BRD to talk about infection of both the upper and lower respiratory tract, and BP to talk about lower respiratory tract infection. I did the modification throughout the text.

Comment 2: Line 38, techniques’ will be more adequate than, tests.

Response 2: Thank you for this suggestion. I did the modification in the text, line 42.

Comment 3: Figure 1 there is no definition for '3'.

Response 3: I thank the reviewer for this comment. The “3” refers to the proximal limit of auscultation’s area. I clarify the legend of the figure 1.

Comment 4: Line 258 please correct, the’ with a lowercase letter.

Response 4: The correction was done.

Comment 5: Table 5, please correct 'consolidation' with a lowercase letter.

Response 5: The correction was done.

Round 2

Reviewer 1 Report

Comments and Suggestions for Authors

I appreciate the authors thorough and complete responses provided. The manuscript provides a compelling overview and achieve its intended purpose.

I only have 2 suggested edits to consider: 

line 67-68: The following short communication 'Short communication: Circadian variations and day to day variability of clinical signs used for the early diagnosis of pneumonia within and between calves' supports the statement.

Line 104: ≥ 3 positive-scoring positive calves? Or is is meant to be '≥ 3 positive-scoring calves'?

Author Response

I thank the reviewer for their relevant comments. Please find included below specific explanations addressing each comment or suggestion made by the reviewer. 

Comment 1: line 67-68: The following short communication 'Short communication: Circadian variations and day to day variability of clinical signs used for the early diagnosis of pneumonia within and between calves' supports the statement.

Response 1: I thank the reviewer for suggesting this interesting reference. I added the reference, line 68.

Comment 2: Line 104: ≥ 3 positive-scoring positive calves? Or is is meant to be '≥ 3 positive-scoring calves?

Response 2: I thank the reviewer for noting this mistake. I reworded the sentence as recommended, line 104.